# Systematic comparison of unilamellar vesicles reveals that archaeal core lipid membranes are more permeable than bacterial membranes

Urszula Łapińska[1], Georgina Glover[1], Zehra Kahveci[1¤], Nicholas A. T. Irwin[2,3], David S. Milner[3], Maxime Tourte[4], Sonja-Verena Albers[4], Alyson E. Santoro[5], Thomas A. Richards[3]*, Stefano Pagliara[1]*

1 Living Systems Institute and Biosciences, University of Exeter, Exeter, United Kingdom, 2 Merton College, University of Oxford, Oxford, United Kingdom, 3 Department of Biology, University of Oxford, Oxford, United Kingdom, 4 Molecular Biology of Archaea, Institute of Biology II, University of Freiburg, Freiburg, Germany, 5 Department of Ecology, Evolution and Marine Biology, University of California, Santa Barbara, California, United States of America

¤ Current address: Instituto de Nanociencia y Materiales de Aragón, CSIC-Universidad de Zaragoza, Zaragoza, Spain

* thomas.richards@biology.ox.ac.uk (TAR); s.pagliara@exeter.ac.uk (SP)

**Citation:** Łapińska U, Glover G, Kahveci Z, Irwin NAT, Milner DS, Tourte M, et al. (2023) Systematic comparison of unilamellar vesicles reveals that archaeal core lipid membranes are more permeable than bacterial membranes. PLoS Biol 21(4): e3002048. https://doi.org/10.1371/journal.pbio.3002048

**Data Availability Statement:** All relevant data are within the paper, its Supporting Information files.

## Abstract

One of the deepest branches in the tree of life separates the Archaea from the Bacteria. These prokaryotic groups have distinct cellular systems including fundamentally different phospholipid membrane bilayers. This dichotomy has been termed the lipid divide and possibly bestows different biophysical and biochemical characteristics on each cell type. Classic experiments suggest that bacterial membranes (formed from lipids extracted from *Escherichia coli*, for example) show permeability to key metabolites comparable to archaeal membranes (formed from lipids extracted from *Halobacterium salinarum*), yet systematic analyses based on direct measurements of membrane permeability are absent. Here, we develop a new approach for assessing the membrane permeability of approximately 10 μm unilamellar vesicles, consisting of an aqueous medium enclosed by a single lipid bilayer. Comparing the permeability of 18 metabolites demonstrates that diether glycerol-1-phosphate lipids with methyl branches, often the most abundant membrane lipids of sampled archaea, are permeable to a wide range of compounds useful for core metabolic networks, including amino acids, sugars, and nucleobases. Permeability is significantly lower in diester glycerol-3-phosphate lipids without methyl branches, the common building block of bacterial membranes. To identify the membrane characteristics that determine permeability, we use this experimental platform to test a variety of lipid forms bearing a diversity of intermediate characteristics. We found that increased membrane permeability is dependent on both the methyl branches on the lipid tails and the ether bond between the tails and the head group, both of which are present on the archaeal phospholipids. These permeability differences must have had profound effects on the cell physiology and proteome evolution of early prokaryotic forms. To explore this further, we compare the abundance and distribution of transmembrane transporter-encoding protein families present on genomes sampled from across

Numerical values for Fig 4 can be found at https://doi.org/10.6084/m9.figshare.22086647.

**Funding:** This work was supported by the Gordon and Betty Moore Foundation Marine Microbiology Initiative (GBMF5514 to TAR., SP and AES), the Biotechnology and Biological Sciences Research Council (BB/V008021/1 to SP and UL), the H2020 Marie Skłodowska-Curie Actions (H2020-MSCA-ITN-2015-675752 to SP and TAR), the Volkswagen foundation (Life? Az 96727 to MT and SVA) and Merton College, University of Oxford (NATI). The funders had no role in study design, data collection and analysis, decision to publish, or preparation of the manuscript.

**Competing interests:** The authors have declared that no competing interests exist,

**Abbreviations:** CF, carboxyfluorescein; G1P, glycerol-1-phosphate; G3P, glycerol-3-phosphate; HMM, hidden Markov model; LUCA, last universal common ancestor; TCDB, Transporter Classification Database.

the prokaryotic tree of life. These data demonstrate that archaea tend to have a reduced repertoire of transporter gene families, consistent with increased membrane permeation. These results demonstrate that the lipid divide demarcates a clear difference in permeability function with implications for understanding some of the earliest transitions in cell origins and evolution.

## Introduction

Membranes are the boundaries that define cells [1]. How metabolites cross membranes is therefore a key factor for understanding early evolution. Early in the tree of life the prokaryotes emerged, cellular forms that include the Bacteria (once called Eubacteria) and the Archaea (once called Archaebacteria) [2–6]. The placement of the root of the prokaryotes is debated [7–9], making it difficult to understand the biology of the last universal common ancestor (LUCA). However, the Archaea/Bacteria bifurcation, which could represent LUCA, marks multiple important differences in cell biology.

Bacteria and Archaea possess homologous ribosomal proteins [6] and ribosomal RNA components [2], yet, Bacteria and Archaea possess distinct cell walls [10], DNA replication machineries (e.g., [11]), DNA packing structures [12], and phospholipid membrane chemistries [13]. This latter distinction, termed the lipid divide [14], is an important evolutionary factor. Archaeal and bacterial phospholipid bilayers are assembled via distinct biosynthetic pathways [7]. Most Bacteria possess membranes of fatty acid chains bonded to a glycerol-3-phosphate (G3P) backbone via ester bonds [7,13,15]. In contrast, mesophilic archaeal membranes are predominantly constructed from diether lipids [16] with isoprenoid chains containing methyl branches bonded to a glycerol-1-phosphate (G1P) backbone via ether bonds [7,13,15] (sometimes called archaeol). These lipids often represent the abundant lipid forms [16], but both Bacteria and Archaea can synthesize a large diversity of structurally distinct lipids often in response to different environments [17]. For example, some Archaea can increase the abundance of bipolar lipids (also known as tetraether lipids or caldarchaeol) in response to high temperatures [18].

The main function of cell membranes is to generate a semipermeable solute barrier that allows for the development of chemical and proton gradients that drive the biochemistry and energetics of life [19]. The core difference between the G3P diester lipids of Bacteria and G1P diether lipids with methyl branches of Archaea could have a profound effect on the primary function of the cell membrane. In vitro studies have identified important biological implications of the physicochemical properties of phospholipid membranes by using liposomes [20–29]. Liposomes are approximately spherical synthetic lipid bilayer membranes with a typical diameter of 100 nm that enclose an internal aqueous phase. A previous study using liposomes suggested that key metabolites from the environment could permeate prebiotically plausible membranes in the absence of transport machinery such as transmembrane transporter proteins [30]. However, this foundational work did not contrast the permeability of core metabolites across archaeal and bacterial-type lipid membranes but rather used mixtures of simple prebiotically plausible lipids, such as fatty acids, fatty alcohols, and monoglycerides. An additional study has shown that liposomes made of diether lipids extracted from archaea (e.g., from *Halobacterium salinarum*) display lower permeability to protons compared to liposomes made of lipids extracted from bacteria (e.g., from *Escherichia coli*) [16]. This feature was confirmed using independent approaches [28,31,32]; the variation in proton permeability being

likely due to the difference in the ether versus ester head–tail bond [33]. This differential permeability constitutes an important trait for understanding the differences in energetic functions of the Archaea and Bacteria [19]. It was also shown that bacterial and diether archaeal lipid membranes display similar permeabilities to glycerol, urea, and ammonia [16]. However, these permeability traits were measured at high extracellular metabolite concentration (i.e., 200 mM) using indirect spectroscopic techniques, which average over a large number of liposomes and associated impurities (e.g., multi-lamellar liposomes and lipid aggregates). As such, there are a number of potential limitations of these results: (i) they are based on ensemble measurements across variant lipid architectural forms; (ii) they do not investigate permeability characteristics of cellular metabolites at biologically/ecologically realistic substrate concentrations; and (iii) they investigate a limited number of metabolites. Collectively, this means that there is a partial understanding of the implications of membrane chemistry variation on metabolite permeability in biologically relevant contexts, particularly given recent evidence suggesting that living cells can obtain nutrients from the environment in a transporter protein–independent fashion [34].

Here, we test the hypothesis that core archaeal and bacterial type lipid membranes have fundamentally different permeability traits, directly implying that the Archaea/Bacteria bifurcation would also encompass a distinct change in metabolite permeability. Such a difference would have profound effects for the evolution of membrane transporter repertoires, intracellular metabolic networks, and associated cellular ecologies. We present a novel approach for the study of membrane permeability based on microfluidic manipulation of unilamellar vesicles composed of a single phospholipid bilayer of archaeal or bacterial type lipids. Our results show that the lipid divide demarcates a dichotomy in membrane permeability characteristics. In contrast to previous ideas, membranes composed of archaeal core phospholipids display elevated permeability to many compounds key for core metabolic functions. Using phylogenomic approaches, we also demonstrate that this functional difference in membrane permeability correlates with variations in the evolution of the transporter protein encoding gene repertoire.

## Results and discussion

### Microfluidic screening to explore membrane permeability characteristics

Here, we report a system to enable the capture and individual placement of unilamellar vesicles obtained via electroformation of synthetic lipids (see Methods) in multiple parallel arrays of tens of vesicles using microfluidics (Fig 1).

This approach enabled us to precisely control the chemostatic fluid environment of the vesicles [35]. The diameter of these vesicles is in the range of 5 to 15 μm. Although such diameter range is large for prokaryotic cell sizes, these dimensions were chosen to aid imaging and manipulation. In order to measure metabolite permeation into the unilamellar vesicles, we loaded the vesicles with a neutral pH buffer and carboxyfluorescein (CF). This fluorescent dye allows direct assessment of metabolite uptake by variations in fluorescent properties in response to changes in intra-vesicle metabolite concentration [16,36]. In fact, the introduction of metabolites in the vesicle reduces the self-quenching properties of CF, resulting in increased vesicle fluorescence. Therefore, a relative increase in intra-vesicle fluorescence indicates membrane permeability to the target metabolite when delivered via continuous flow through the microfluidic device (Fig 1 and Methods). Individual metabolites were delivered into the extra-vesicle environment (also containing a neutral pH buffer) at a concentration of 1 mM, while imaging the changes in fluorescence levels of multiple individually trapped vesicles (Fig 1 and Methods). By using this experimental approach, we conducted parallel controlled experiments exploring how cellular metabolites can cross membranes of different phospholipid chemical

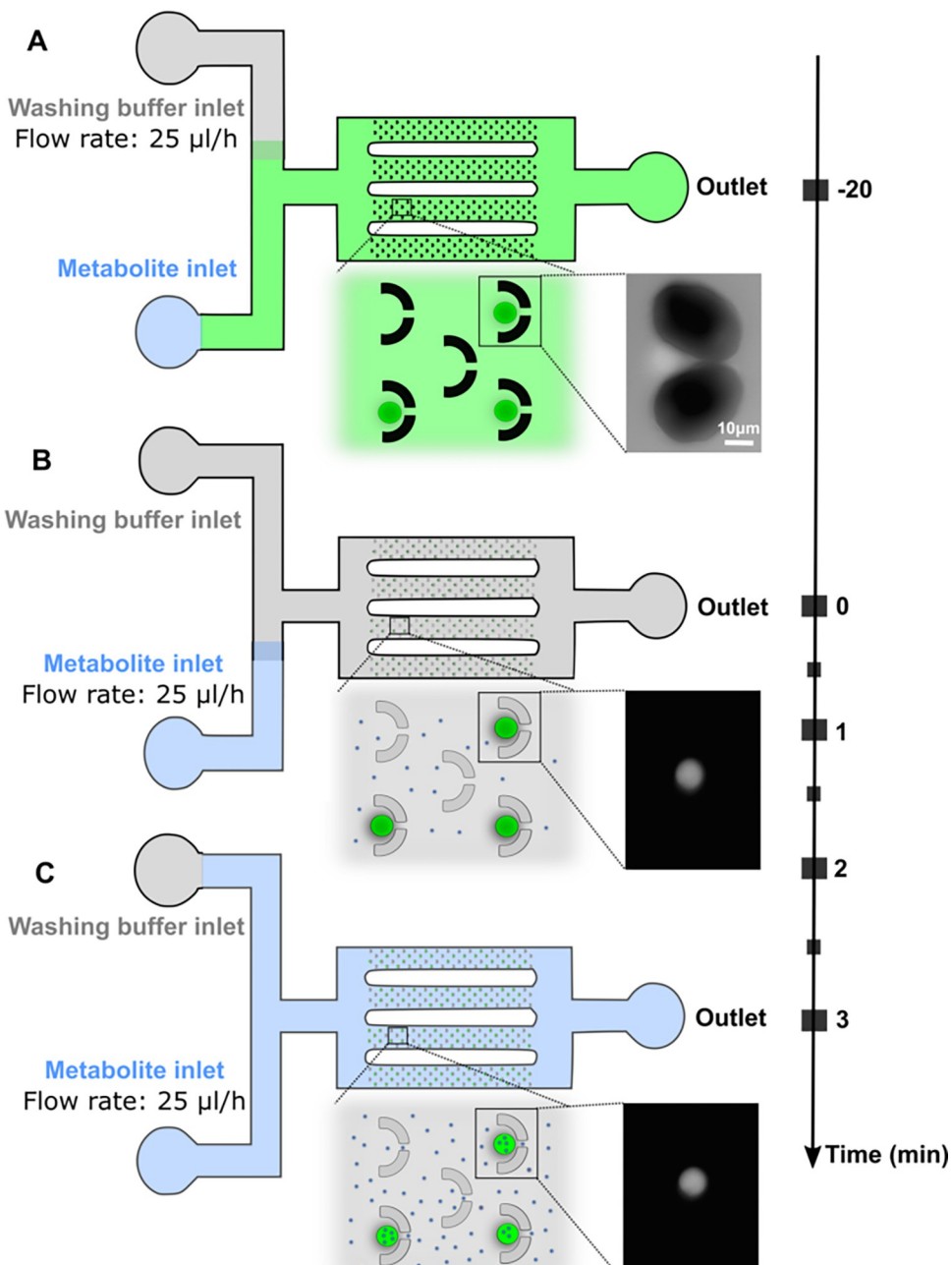

**Fig 1. Schematics illustrating the microfluidic approach used to study membrane permeability. (A)** Vesicles (green circles) loaded into the fluidic device (at t = −20 min) display an intracellular carboxyfluorescein (CF) concentration that is similar to the extracellular CF concentration; thus, vesicles confined in the fluidic coves appear as bright as the extracellular environment, as illustrated by the micrograph. Free CF molecules are removed from the microfluidic environment around the vesicles by applying a flow rate of 25 μL/h from the washing buffer inlet for 20 min. **(B)** At t = 0 min, vesicles appear bright on a dark background as shown in the micrograph. The metabolite (small blue circles) under investigation is delivered to the vesicles by applying a flow rate of 25 μL/h from the metabolite inlet for 3 min. **(C)** The metabolite accumulates within the vesicles if their membrane is permeable to the metabolite.

composition; we used relative changes in CF fluorescence as a reporter for relative permeability to each metabolite of the different lipid membranes investigated.

We first studied the permeability of single vesicles composed of synthetic lipids with isoprenoid chains containing methyl branches bonded to a G1P backbone via ether bonds (archaeal-like membrane phospholipids—abbreviated here as "archaeal 4ME diether G1PC") (lipid 1 in S1 Table) or synthetic lipids with fatty acids bonded to a G3P backbone via ester bonds (bacterial-like membrane phospholipids—ternary lipid mixture abbreviated here as "bacterial diester G3PE-PG-CA", lipid 2 in S1 Table) to 18 small metabolites (S2 Table). Notably, synthetic G1P diether lipids with methyl branches are not commercially available, to our knowledge, and were therefore synthesised de novo for the purpose of this study (see Methods). We chose metabolites with different molecular weight, hydrophobicity, and number of rotatable bonds (S2 Table).

Our single-vesicle measurements revealed heterogeneity in the permeability of each synthetic lipid type to each metabolite: Some vesicles of each lipid type displayed a decrease in intracellular fluorescence during the delivery of each metabolite, while other vesicles displayed an increase in intracellular fluorescence (temporal dependence of single-vesicle fluorescence for archaeal 4ME diether G1PC lipids and bacterial diester G3PE-PG-CA lipids are reported with dashed magenta lines and dashed-dotted blue lines in S1 Fig and Data A in S1 File). For example, the coefficient of variations in the permeability of archaeal 4ME diether G1PC vesicles and of bacterial diester G3PE-PG-CA vesicles to aspartic acid, glyceraldehyde, and adenine were 118% and 103%, 60% and 198%, 76% and 108% (S1D, S1G and S1Q Fig, respectively), in accordance with the proposition that lipid membranes are a system with heterogeneous functions [37–42].

Therefore, in order to compare the permeability of different synthetic lipid types to the same metabolite, we carried out Mann–Whitney two-tailed statistical comparisons between the distributions of single intra-vesicle fluorescence after 3 min of delivery of each metabolite for each synthetic lipid type. Temporal dependence of fluorescence distribution means and standard deviations are reported in Fig 2 (magenta triangles and dashed lines for archaeal 4ME diether G1PC vesicles, blue squares and dashed-dotted lines for bacterial diester G3PE-PG-CA vesicles) together with single-vesicle fluorescence distributions and Mann–Whitney two-tailed statistical comparisons at t = 3 (further details about Mann–Whitney two-tailed statistical comparisons are reported in S2 File).

We found that archaeal 4ME diether G1PC vesicles were significantly more permeable to the amino acids glycine, alanine, leucine, aspartic acid, tryptophan, and glutamine compared to bacterial diester G3PE-PG-CA vesicles (Figs 2A–2F and S1A-S1F and Data A in S1 File). Consistent with the amino acid findings, archaeal 4ME diether G1PC vesicles were also more permeable to the sugars glyceraldehyde, glycerol, deoxyribose, ribose, and arabinose compared to bacterial diester G3PE-PG-CA vesicles (Figs 2G–2K and S1G-S1K and Data A in S1 File), whereas we did not measure a significant difference in the permeability to dihydroxyacetone (Figs 2L and S1L and Data A in S1 File). Interestingly, the difference in permeability was strongly distinct for three relatively large sugar types, deoxyribose, ribose, and arabinose, the two former sugars including primary constituents of the hereditary materials DNA and RNA, respectively.

Finally, archaeal 4ME diether G1PC vesicles were also more permeable to the amide urea, and the nucleobases cytosine, uracil, guanine, and adenine compared to bacterial diester G3PE-PG-CA vesicles (Figs 2M–2Q and S1M-S1Q and Data A in S1 File). These data suggest that important nitrogen sources and components of DNA and RNA can permeate archaeal 4ME diether G1PC vesicles. The phosphonate 2-aminoethyl phosphonic acid (Figs 2R and S1R and Data A in S1 File) showed no significant difference in permeability characteristics between the two types of vesicles.

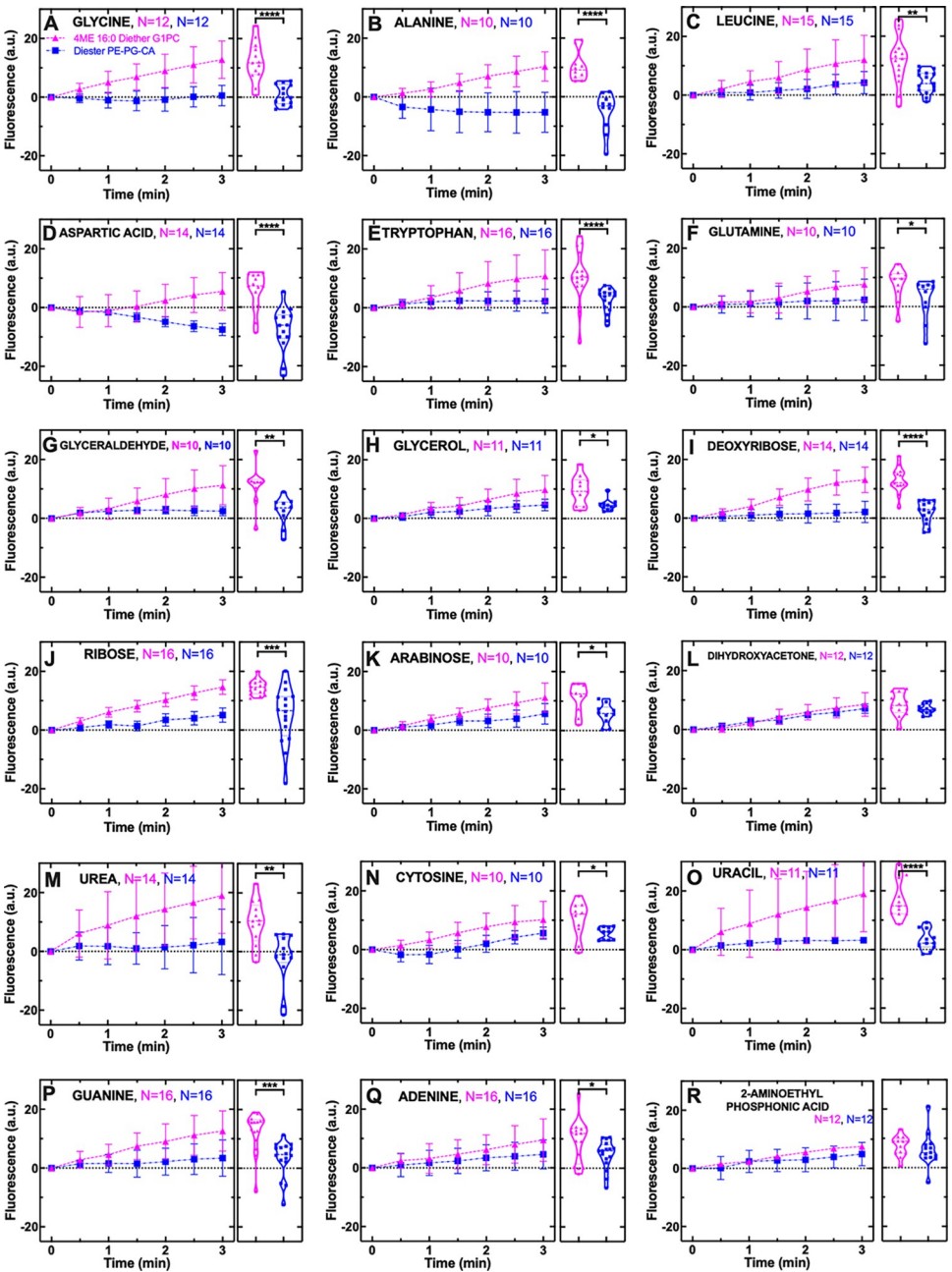

**Fig 2. Archaeal 4ME diether G1PC vesicles are consistently more permeable to a range of metabolites than bacterial diester G3PE-PG-CA vesicles.** Temporal dependence of average CF fluorescence in archaeal 4ME diether G1PC (magenta triangles) and bacterial diester G3PE-PG-CA vesicles (blue squares) during the exposure to 1 mM of variant metabolites delivered to the microfluidic coves. Mean (symbols) and standard deviation (error bars) were calculated from at least 10 single-vesicle measurements across three independent experiments. Lines are guides for the eye. N is the number of single vesicles investigated for each metabolite exposure and each type of vesicles (magenta and blue for archaeal 4ME diether G1PC and bacterial diester G3PE-PG-CA vesicles, respectively). N varies across different metabolite experiments investigated due to technical constraints (see Methods). However, care has been taken to obtain the same N for each metabolite experiment across the two different types of vesicles to ensure reliable statistical comparisons. Such comparisons have been carried out via two-tailed Mann–Whitney tests between the distributions of CF fluorescence values at t = 3 min for archaeal 4ME diether G1PC and bacterial diester G3PE-PG-CA vesicles. ****: $p$-value $< 0.0001$, ***: $p$-value $< 0.001$, **: $p$-value $< 0.01$, *: $p$-value $< 0.05$ and are shown with corresponding violin plots next to each time-course graph. Numerical values of CF fluorescence in individual vesicles for each lipid type during the delivery of each metabolite are provided in Data A in S1 File. The lipids used for creating archaeal 4ME diether G1PC and bacterial diester G3PE-PG-CA vesicles are lipids 1 and 2, respectively, in S1 Table.

Pooling together the data for the 18 metabolites investigated, we did not find a significant correlation between permeability and lipid molecular weight or hydrophobicity (Pearson coefficient r = −0.22, 0.36, respectively, for archaeal 4ME diether G1PC vesicles; r = 0.09, 0.24, respectively, for bacterial diester G3PE-PG-CA vesicles; none were significant at $p$ = 0.05; S2 Table). These data suggest that the metabolic selectivity of these membrane mimics is complex and does not rely solely on basic molecular properties, such as molecular weight or hydrophobicity. Taken together with previous findings, demonstrating that solute hydrophobicity correlates poorly with permeability coefficients of eukaryotic fatty acid or phospholipid membranes [36] and that amino acid residue hydrophobicity might have not been key for the emergence of the genetic code [43], our data corroborate the hypothesis that subtle variations in the metabolite atomic structure could contribute to differences in membrane permeability [36,37,44]. In fact, we found a significant negative correlation between permeability and rotatable bond number for archaeal 4ME diether G1PC vesicles (Pearson coefficient r = −0.49, *), but not for bacterial diester G3PE-PG-CA vesicles (r = −0.32, nonsignificant at $p$ = 0.05). It is also conceivable that other physicochemical properties such as the partition energy of amino acids (i.e., the propensity of amino acid residues to be exposed to water [45]), which has recently been proposed to be the most optimized property of amino acids in the genetic code [46], could contribute to differences in membrane permeability.

The striking difference in CF fluorescence between archaeal 4ME diether G1PC and bacterial diester G3PE-PG-CA vesicles was confirmed when we extended the duration of our permeability experiments from 3 to 6 min (S2 Fig and Data A in S1 File). Moreover, archaeal 4ME diether G1PC and bacterial diester G3PE-PG-CA vesicles displayed similar dimensions in range 5 to 15 μm, and the dimensions of each vesicle did not significantly change during metabolite delivery (S3 Fig and Data A in S3 File). These data confirm that vesicle deformation did not occur in our experiments and that neither differences in vesicle curvature nor deformation play a role in the observed differences in permeability traits between archaeal 4ME diether G1PC vesicles and bacterial diester G3PE-PG-CA vesicles. We cannot exclude, however, the possibility that differences in curvature between bacteria and archaea, with dimensions in range 1 to 5 μm, could play a role in permeability to metabolites.

Such differences were also not attributable to differential metabolite accumulation within the lipid bilayers, since (i) CF has very low affinity for the lipid hydrophobic chains [16], so interactions with substrates within the membrane are unlikely; (ii) CF fluorescence intensity was uniform across whole vesicles for both lipid types and the full range of metabolites investigated, consistent with previous data obtained on fatty acid and phospholipid liposomes [16,36]. Our data do not allow us to infer detailed kinetics of the permeation of each metabolite and could not be complemented via pulse-chase experiments requiring faster fluidic exchanges. Therefore, we did not attempt to extract absolute kinetic parameters, such as the permeability coefficient, but centred this current work on directly comparing relative changes in CF fluorescence (as a proxy for permeability) between archaeal 4ME diether G1PC and bacterial diester G3PE-PG-CA vesicles. However, we were able to observe differences in terms of both the uptake onset and slope for different metabolites. For example, glycine, ribose, and uracil displayed a steep uptake during the first minute of their delivery to archaeal 4ME diether G1PC vesicles (Fig 2A, 2J and 2O, respectively), whereas the uptake of aspartic acid, glutamine, and dihydroxyacetone started only after the first minute of their delivery to archaeal 4ME diether G1PC vesicles (Fig 2D, 2F and 2L). These data demonstrate that metabolites are not passing across vesicles via puncture holes generated during vesicle formation but by genuine diffusion through the lipid bilayers in a metabolite-specific manner.

As discussed in the introduction, natural archaeal membranes are formed from heterogenous mixtures of lipids some with tetraether bipolar lipids (e.g., caldarchaeol), which act to

directly connect the membrane bilayers, a function that is likely to increase the stiffness of the membrane and reduce permeability [47]. Such mixtures might have different properties than the homogenous membranes studied here. To explore this possibility, we attempted to use lipids extracted from *Haloferax volcanii*, predominantly containing diether lipids with head group derivatives of phosphatidylglycerol [48], and lipids extracted from *Sulfolobus acidocaldarius*, predominantly containing bipolar tetraether lipids with head group derivatives of phosphatidylhexose [49]. However, we could not obtain mechanically stable vesicles via electroformation for either of these lipid mixtures. It is therefore important to mention that our experiments do not reveal the permeability traits of extant prokaryotic membrane mixtures but rather identify the contrasting permeability traits of the common and core building blocks of the archaeal and bacterial membranes.

Finally, we wanted to rule out that the relatively lower permeability of bacterial diester G3PE-PG-CA vesicles could be due to interactions between different lipids within the ternary lipid mixture that we employed to mimic more closely bacterial membranes (lipid 2 in S1 Table). To do so, we measured and contrasted permeability to urea, glycine, ribose, deoxyribose, glycerol, and phosphonate in vesicles made of bacterial ternary-lipid mixtures (G3PE-PG-CA, lipid 2 in S1 Table) and vesicles made of single lipids (G3PE, lipid 6 in S1 Table) and found that these two different bacterial mimics displayed comparably low permeabilities to all the metabolites tested (S4 Fig and Data B in S1 File). We selected these six metabolites because archaeal 4ME diether G1PC vesicles and bacterial diester G3PE-PG-CA vesicles display different patterns of permeabilities to these metabolites (Fig 2).

## Which archaeal lipid characteristics determine permeability traits?

To uncover the chemical determinants of archaeal membrane permeability, we employed vesicles made of a range of lipids with a mixture of archaeal and bacterial lipid characteristics. We tested the impact of the lipid chain branching, length, tail–head bond (ester/ether), and the G1P versus G3P backbone on membrane permeability. As above, we performed these experimental tests using urea, glycine, ribose, deoxyribose, glycerol, and phosphonate.

Considering that the G1P backbone is one of the most characteristic traits of archaeal lipids [50–53], we firstly electroformed vesicles by using a lipid that carried a bacterial-like G3P backbone but with archaeal like diether tail–head bond and isoprenoid chains containing methyl branches (4ME diether G3PC, lipid 3 in S1 Table, green circles in Fig 3 and Data C in S1 File). The permeabilities of these hybrid vesicles to the six metabolites investigated were not significantly different to the permeabilities of archaeal vesicles (4ME diether G1PC, lipid 1 in S1 Table, magenta upward triangles in Fig 3, Kruskal–Wallis one-way analysis of variance statistical comparisons are reported in S4 File) but were significantly higher than the permeabilities measured for the bacterial vesicles (diester G3PE-PG-CA, lipid 2 in S1 Table, blue squares in Fig 3 and S4 File). These data suggested that the change from a G1P to a G3P backbone is not a key factor in determining membrane permeability.

Next, we tested a different hybrid lipid with the bacterial-like G3P backbone and lipid chains without methyl branches, but with archaeal-like diether tail–head bond (diether G3PC, lipid 4 in S1 Table and black diamonds in Fig 3). We found that in the absence of lipid chain branching, these hybrid vesicles displayed a statistically significant and consistently lower permeability compared to archaeal 4ME diether G1PC vesicles (magenta upward triangles in Fig 3 and S4 File). This striking difference is possibly due to increased membrane fluidity in the presence of methyl branched lipid chains [30,54]. In the absence of methyl branching, these hybrid diether G3PC vesicles displayed permeabilities to urea, glycine, deoxyribose, or ribose that were comparable to the permeabilities measured for bacterial diester G3PE-PG-CA

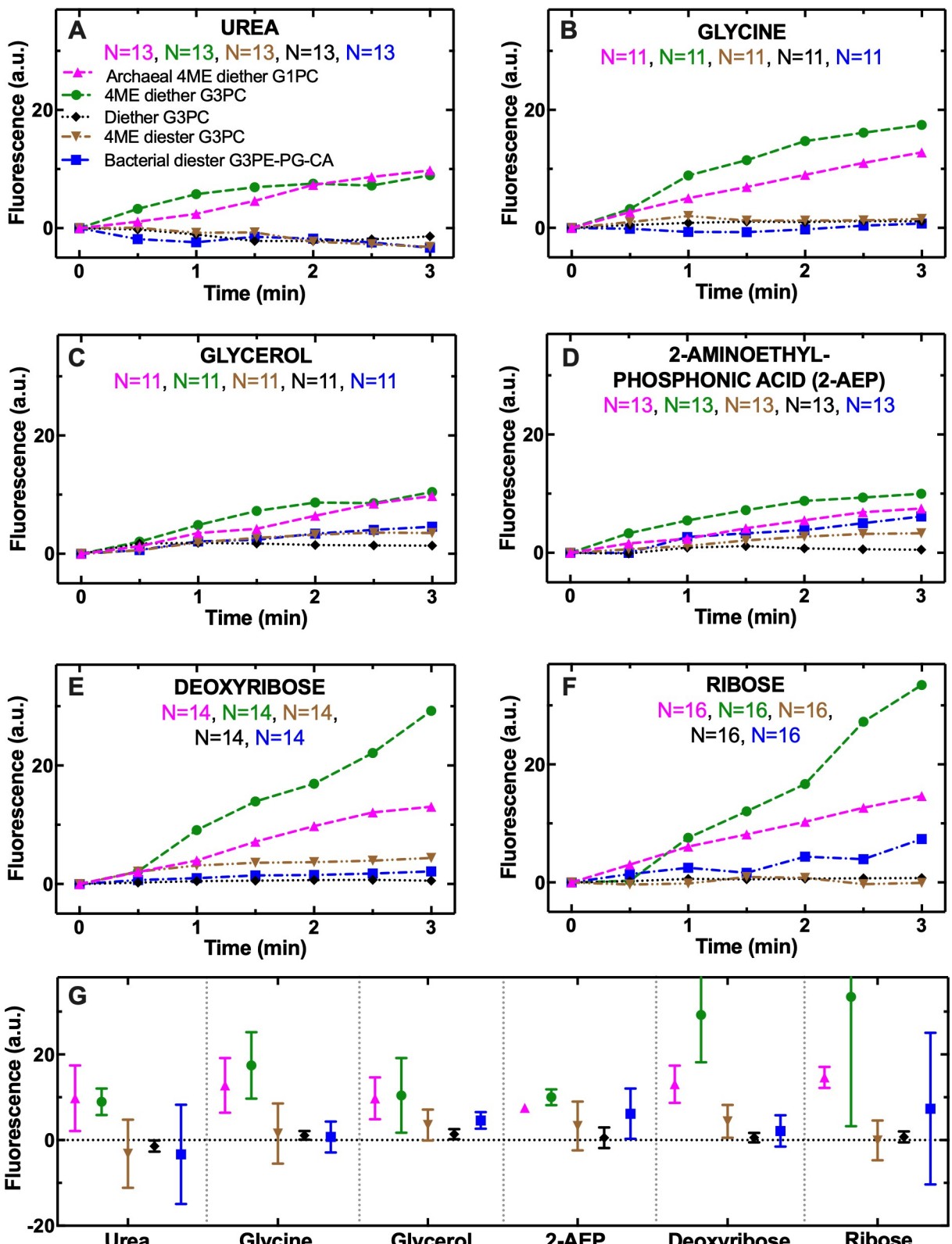

**Fig 3. Chain branching and ether bond have profound implications on archaeal membrane permeability to small metabolites.** Temporal dependence of average CF fluorescence in archaeal 4ME diether G1PC (magenta upward triangles), bacterial diester G3PE-PG-CA (blue squares), 4ME diether G3PC (green circles), 16:0 Diether G3PC (black diamonds), or 4ME 16:0 Diester G3PC (brown downward triangles)

vesicles during the exposure to 1 mM (**A**) urea, (**B**) glycine, (**C**) glycerol, (**D**) 2-aminoethyl phosphonic acid, (**E**) deoxyribose, or (**F**) ribose delivered to the vesicles at t = 0. These substrates were chosen as previous analyses demonstrate that these had key differences in permeability characteristics. Data for archaeal 4ME diether G1PC and bacterial diester G3PE-PG-CA vesicles are reproduced from Fig 2 for comparison purposes. Means (symbols) were calculated from at least 10 single-vesicle measurements across 3 independent experiments. Due to the large overlaps between the measurements obtained with the different vesicle types, standard deviations and single-vesicle measurements are not reported for clarity. Lines are guides for the eye. Mean and standard deviations measured at t = 3 min and statistical comparisons carried out via Kruskal–Wallis one-way analysis of variance between the distributions of CF fluorescence values at t = 3 min for each type of vesicle pair are reported in (**G**) and in S1 File. ****: *p*-value < 0.0001, ***: *p*-value < 0.001, **: *p*-value < 0.01, *: *p*-value < 0.05. N is the number of single vesicles investigated for each metabolite and each lipid type. N varies across different metabolite experiments investigated due to technical constraints (see Methods). However, care has been taken to obtain the same N for each metabolite experiment across each pair of lipid type to ensure reliable statistical comparisons. Mean and standard deviation of numerical values of CF fluorescence in individual vesicles for each lipid type during the delivery of each metabolite are provided in Data C in S1 File. The lipids used for creating archaeal 4ME diether G1PC, bacterial diester G3PE-PG-CA, 4ME diether G3PC, 16:0 Diether G3PC, or 4ME 16:0 Diester G3PC vesicles are lipids 1, 2, 3, 4, and 5, respectively, in S1 Table.

vesicles (lipid 2 in S1 Table and blue squares in Fig 3A, 3B, 3E and 3F, respectively, none significantly different at *p* = 0.05; S4 File). In contrast, permeabilities to glycerol and phosphonate were lower in the hybrid diether G3PC vesicles without methyl branching compared to bacterial diester G3PE-PG-CA vesicles (Fig 3C and 3D, * and *, respectively; S4 File).

A third hybrid lipid with the bacterial-like G3P backbone and a diester tail–head bond but with archaeal-like isoprenoid chains with the methyl branches was also compared in an equivalent manner (4ME Diester G3PC, lipid 5 in S1 Table, brown downward triangles in Fig 3). In the absence of an ether bond (substituted with an ester bond), 4ME diester G3PC vesicles displayed significantly lower permeabilities to all six metabolites investigated, compared to archaeal 4ME diether G1PC vesicles (magenta upward triangles in Fig 3 and S4 File), and exhibited permeabilities similar to bacterial diester G3PE-PG-CA vesicles (blue squares in Fig 3, none significantly different apart from permeability to ribose that was significantly lower in 4ME Diester G3PC vesicles, *; S4 File). Taken together, these data demonstrate that increased membrane permeability can be achieved via the simultaneous use of an ether bond and methyl chain branching, both of which characterize the core lipids of archaea.

To follow on from this, we set out to determine whether permeability is affected by variations in the archaeal lipid head (lipid 7 in S1 Table, with phosphoethanolamine instead of phosphocoline lipid head). However, despite attempting different electroformation protocols (see S3 Table), we could not produce vesicles using this lipid, possibly because this lipid forms non-lamellar structures of either a cubic or hexagonal fashion [55]. We note, however, that previous studies suggest that lipid headgroup composition has only limited impact on the permeation of small molecules [29].

Next, we investigated how permeability varies in phospholipids according to chain length. Synthetic G1P lipids with methyl branches are not commercially available, so we focused on the study of G3P lipids without methyl branches and with variant chain lengths (lipids 4, 8, and 9 in S1 Table). These experiments provide no evidence for a significant correlation between lipid chain length (C = 12, C = 16, and C = 18, i.e., lipids 8, 4, and 9 in S1 Table) and permeability to any of the metabolites investigated (S5 Fig and Data D in S1 File, Pearson correlation coefficient r = −0.58, 0.95, 0.52, −0.10, −0.69, and 0.30 for urea, glycine, glycerol, phosphonate, deoxyribose, and ribose, respectively), corroborating previous findings on eukaryotic membrane mimics [56]. Significant differences in permeability to water and weak acids were previously observed only for large variations in chain length; an increase from 14 to 26 in the length of acyl chains of eukaryotic lipids led to a 5-fold decrease in permeability [29]. Accordingly, our data show that a short chain length slightly favoured permeability to urea and deoxyribose (S5A and S5E Fig, respectively). In contrast, a long chain length slightly favoured permeability to glycine and ribose (S5B and S5F Fig, respectively), but overall, these effects were masked by vesicle-to-vesicle variation in permeability to these metabolites. Furthermore,

we attempted to produce vesicles using lipids of other chain lengths (C = 6 and C = 14, i.e., lipids 10 and 11 in S1 Table). However, these vesicles appeared to be mechanically unstable, possibly because their transition temperature is close to the temperature at which we carried out our membrane permeability assays. We note that these comparisons did not include variations in the number of methyl branches per chain, so our experiments do not rule out the possibility that differences in the number of methyl branches may alter permeability characteristics.

Our next experiments demonstrated that decreasing bonding saturation (i.e., single bonds that were more likely present in prebiotic molecules [30] versus double bonds) along hybrid G3P diether phospholipids of fixed chain length (C = 18, lipid 9 in S1 Table) significantly decreased permeability to the small amide, urea, and, to a lesser extent, to the small amino acid, glycine, compared to bonding unsaturation (i.e., double bonds, lipid 12 in S1 Table, S6A and S6B Fig, and Data E in S1 File, *** and *, respectively). However, bonding saturation did not have an impact on the permeability to glycerol, phosphonate, deoxyribose, or ribose (S6C, S6D, S6E and S6F Fig, respectively, none significantly different), whereas previous studies using G3P diester phospholipids or fatty acid liposomes found increased permeability to glycerol [29] and ribose [30] in liposomes made of unsaturated lipids.

Tetraether bonds, generating bipolar lipids (or caldarchaeol), or cyclopentane rings along the caldarchaeol chains could further affect the permeability traits in archaeal membrane mimics. Given the variance of these lipid forms across Archaea [57] and the observation that such variants are the minority constituents in some archaeal membranes [58,59], we suggest they are of lesser importance for understanding the evolution of core permeability functions in ancestral cell forms carrying archaeal lipids. However, we do acknowledge that introduction of these lipid variations is likely to alter permeability [60] and is an important factor for determining the ecology of extant Archaea including any ancestral archaeal evolutionary bottleneck that included obligate thermophilic ecology [61]. The experimental platform presented is readily adaptable to investigate the effect of further chemistry variations; however, these synthetic lipids are not currently commercially available. Future work should explore the effect of lipid mixtures on permeability traits. Furthermore, we did not investigate the effect on permeability of membrane variants embedding an S-layer, a peptidoglycan layer, or an outer membrane such as those seen in diderms, because all these features are predicted to follow the archaeal/bacterial bifurcation and so represent secondary elaborations relative to the lipid divide. We also acknowledge that archaea inhabit a variety of environments: Some species thrive at ambient temperature and at neutral pH, whereas others inhabit more "hostile" environments with variations in pH, temperature, or salinity levels [62,63]. Permeability traits may vary with changes in environmental conditions such as pH (which we kept constant at 7.4), temperature (which we kept constant at 22°C) or salinity levels [64,65]; therefore, the permeability traits we have observed are likely subject to variation in different environmental conditions. Although our experimental platform could be adapted to investigate permeability traits under variant environmental conditions, there are constraints as vesicles suffer damage in the presence of very low or very high pH and/or salinity. Moreover, lipids change state above or below the transition temperature. Therefore, the experimental platform would need further development to physically stabilise vesicles, by using, for example, higher density media or by forming vesicles on physical support structures. However, such experiments would tell us much about the conditions in which cellular chasses evolved.

## Membrane permeability negatively correlates with transporter gene repertoires

The observed differences in membrane permeability imply that any transition between archaeal and bacterial-type lipid membrane chemistries would require extensive recalibration

of numerous cellular systems in response to changes in permeability, osmotic stress, and metabolite homeostasis. Such a transition could be facilitated by a mixed archaeal–bacterial lipid membrane [7,66], but the ultimate change would require adaptation to multiple altered cellular properties. The evolution of membrane transporters could permit a reduction in lipid membrane permeability, for example, if a cell was to transition from an archaeal-like lipid membrane to a bacterial-like lipid membrane, as predicted under some hypotheses for the origins of the eukaryotes [67,68]. Accordingly, given the increased permeability of the core archaeal lipid membranes shown here, we hypothesized that archaeal genomes would encode a significantly reduced complement of transporter gene families relative to Bacteria, particularly for those protein families known to transport metabolites capable of permeating archaeal lipid membranes (shown in Fig 2). A limited transporter repertoire could reflect a reduced dependency on protein-based translocation systems as metabolite requirements could be satisfied by a combination of core metabolic function (autotrophy) and lipid membrane permeability. Likewise, increased membrane permeability may limit the utility of membrane transporters by decreasing transport efficiency or impairing the formation of concentration gradients.

To test this hypothesis, we iteratively searched diverse bacterial ($n$ = 3,044) and archaeal ($n$ = 243) genome-derived predicted proteomes using profile hidden Markov models (HMMs; $n$ = 277) derived from TCDB (Transporter Classification Database) protein families [69], to identify previously classified transporter homologs across prokaryotes. Despite the sensitivity of our search, the Archaea had fewer transporters relative to the Bacteria, irrespective of bacterial membrane system (e.g., monoderms or diderms) and regardless of whether transporter numbers were normalised to the total number of proteins per genome (Figs 4A and S7A).

Certain transporter families were consistently encoded across prokaryotes, including many ion transporters (see cluster 2 in Fig 4A, largely composed of ion transporters, for example), which are required due to the impermeability of both membrane types to ions, with the exception of protons [56]. In contrast, other families showed significantly reduced representation in Archaea (see clusters 1, 3, and 5 in Fig 4A, which were functionally heterogeneous, whereas cluster 4 comprised outer membrane transporters associated with gram-negative bacteria). In particular, transporter families known to translocate metabolites similar to those that permeate the archaeal type lipid membrane (e.g., amino acids, sugars, and nucleobases shown in Fig 2) were significantly depleted even when accounting for differential taxon sampling bias using bootstrap resampling (Fig 4B). Nonetheless, there is a significant bias towards bacterial genome sampling in these datasets, and further characterization of additional archaeal genomes will be important for fully reconstructing archaeal transporter repertoires and further testing the trends identified here.

It has previously been argued that protein–membrane interactions, attuned to the specific lipid characteristics of either bacterial or archaeal membranes, may have acted to enforce the lipid divide as the Bacteria and Archaea diversified [14]. Therefore, to account for the possibility that archaeal transporters were not accurately recovered in our searches due to divergent biochemical characteristics or a lack of archaeal transporter family representation in TCDB, we first examined the possibility that archaeal membrane transporters have transmembrane domains with different lengths or alternative amino acid compositions, a factor that may have obscured previous HMM-based annotations of transporter gene families. Comparisons between over 10.8 million bacterial and 528,000 archaeal TM domains (identified from over 2.2 million bacterial and 116,000 archaeal transporter proteins) revealed no significant differences in TM domain characteristics between Archaea and Bacteria (S7B and S7C Fig), suggesting that these biochemical properties would neither impair bioinformatic detection nor would limit horizontal gene transfer of transporter proteins from Bacteria to Archaea [14]. However,

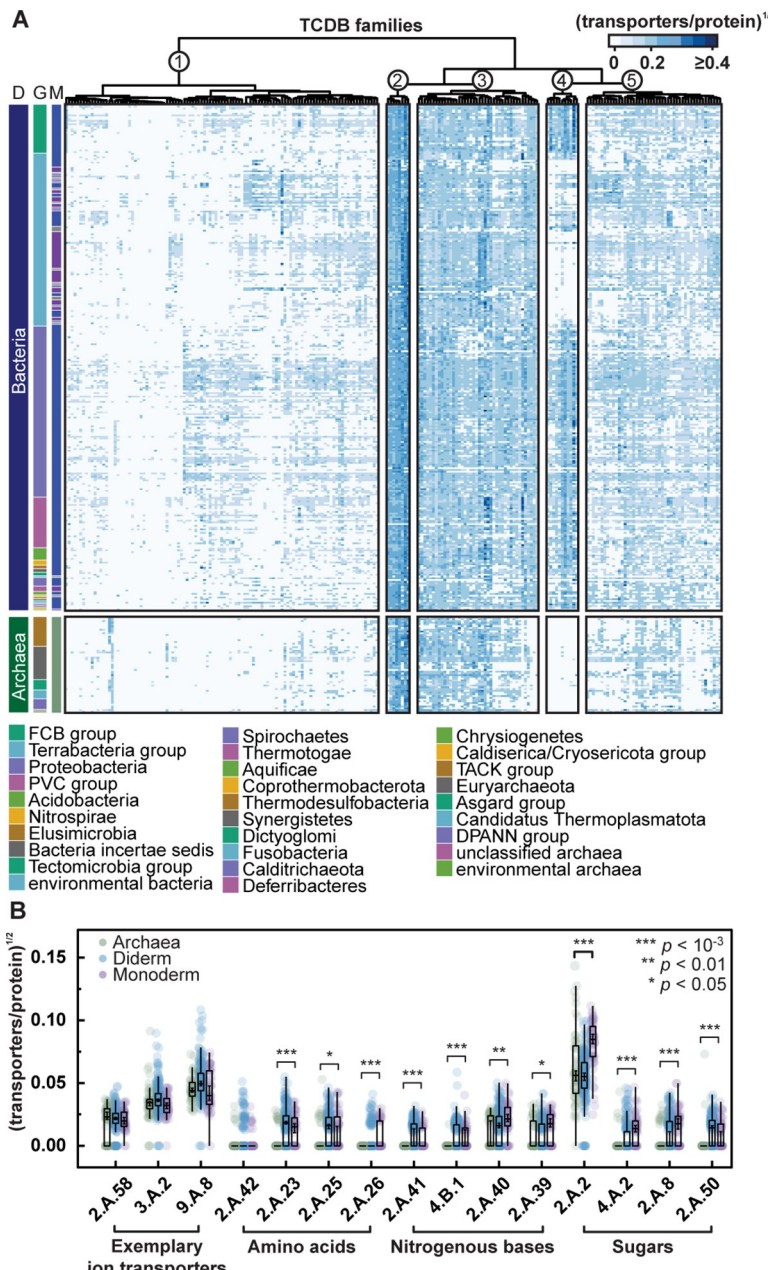

**Fig 4. Archaea feature reduced transporter repertoires relative to Bacteria. (A)** Heat map showing transporter repertoires in Archaea and Bacteria, where each row and column represent a prokaryotic order and TCDB transporter family, respectively. Heat map values represent the median number of transporters normalized by total protein count for every taxon across each prokaryotic order after a square root transformation to account for bias sampling of some taxa. The domain (D), group (G), and membrane morphology (M: where blue, purple, and grey represent diderms, monoderms, and unknown membrane morphology, respectively) of each order is noted. TCDB families were grouped by hierarchical clustering based on transporter abundance using Euclidean distances and the Ward.D2 clustering method. Taxonomy was based on NCBI Taxonomy classifications. **(B)** Individual comparisons of transporter families in Archaea, monoderms, and diderms. All transporter families predicted to translocate amino acids, nitrogenous bases, and sugars are shown, alongside three exemplary ion transporter families from Cluster 2. Comparisons were made using Wilcoxon tests, and Bonferroni-corrected $p$-values are shown. To assess taxon sampling bias, bootstrap-estimated 90% confidence intervals ($n = 1,000$) of the median are displayed over each boxplot median based on random sampling of prokaryotic orders with replacement. Numerical values of the data presented are provided at https://doi.org/10.6084/m9.figshare.22086647.

we note that differences may be obscured because of lower archaeal sampling and absence of systematic functional studies of archaeal transporters not identified by cross-referencing bacterial transporter families.

Secondly, to identify putative archaeal transporters that may have been absent from the TCDB database, we clustered archaeal proteins into protein families and identified those with characteristics indicative of transporter proteins (a median of at least four TM domains and annotation with transporter-associated PFAM domain). This search revealed only 13 previously unsampled putative transporter families (mostly branched-chain amino acid permeases, ABC- and EamA-transporters) that were predicted to function in metabolite uptake and were present in at least a quarter of archaeal species (S5 File). However, only a single undetected putative ABC-transporter family was identified in over 50% of archaeal species (70.8%). These results strongly indicate that unidentified transporter gene families do not account for the reduced transporter repertoire observed in Archaea (S5 File). Together, these results are consistent with a reduced dependency on metabolite transporter proteins in Archaea, an evolutionary outcome that could reflect a reduced transporter repertoire in the last archaeal common ancestor or frequent transporter losses in individual archaeal lineages. Regardless, transporter reduction may have been facilitated, in part, by the increased metabolite permeability of archaeal lipid membranes, which, in turn, reduces the requirement for, or utility of, transporter proteins. However, it is also conceivable that if Archaea were more autotrophic than Bacteria, there may have been a greater tendency to lose membrane transporters in Archaea compared to Bacteria considering the increased propensity of transporter differential loss compared to water-soluble proteins [70].

## Conclusions

The branching point between the Archaea and the Bacteria is a candidate for the deepest branch in the tree of life, which can be conceivably reconstructed using comparative biology and phylogenomic methods. This ancient node demarks two different core membrane lipid chemistries and is known as the lipid divide. There is considerable variation in membrane lipid composition on both sides of the divide, but fundamentally, these membranes are built of two different core phospholipid chemistries.

Here, we demonstrate that homogenous versions of the archaeal and bacterial core lipid membranes show distinct differences in permeability characteristics when generated using vesicle electroformation. Archaeal-type lipid vesicles show permeability to a range of compounds that would theoretically be useful to known cellular metabolic systems and therefore putative proto-metabolic networks. Our data demonstrate that archaeal-type lipid vesicle permeability is dependent on the simultaneous presence of methyl chain branching and ether bond properties, two hallmarks of archaeal lipids. This selective permeability could potentially have provided the lineage that became the Archaea with access to specific metabolic resources without the need for a diversified cross-membrane transporter system. Such an evolutionary scenario suggests that the early archaeal forms and possibly early proto-prokaryotic forms with an archaeal-type core membrane (before the membrane transition, which marks the Archaea–Bacteria bifurcation and indeed the evolution of membrane transporters) possessed many heterotrophic functions (i.e., the ability to acquire chemical precursors for metabolic function via permeation from an external environment) [30,71,72]. Such a scenario does not exclude an early dependency on autotrophic functions at the origins of life [19,43,73,74] or indeed a metabolic network largely reliant on autotrophic function as the Archaea arose [9,61], nor does it exclude early phases of cellular evolution based on fatty acid membranes [30,75]. However, our results suggest as early life transitioned to prokaryote-type cells the

acquisition of a proto-metabolic network within a prokaryotic membrane chassis could more readily be achieved within a core archaeal-type membrane chemistry. We argue that this is logical, because even if a central autotrophic proto-metabolism was entirely self-sufficient, as this system was expanded towards wider functions seen in LUCA, the ability to acquire and integrate additional metabolic resources, such as diversified nucleobases and amino acids, would be feasible in an archaeal chassis. This is because as proto-metabolic networks were compartmentalised and diversified within prokaryotic-like membranes, archaeal chassis possessed permeability to metabolic precursors, whereas bacterial type membranes did not.

## Methods

### Preparation of materials

All chemicals were purchased from Merck. All lipids [i.e., lipid 1 (1,2-di-O-phytanyl-sn-glycero-3-phosphocholine), lipid 2 (1,2-dioleoyl-sn-glycero-3-phosphoethanolamine, 1,2-dioleoyl-sn-glycero-3-phospho′ (1′-rac-glycerol), 1,3-*bis* ('*n*-3′-phosphatidyl)-*sn*-glycerol), lipid 3 (1,2-dioleoyl-sn-glycero-3-phosphoethanolamine), lipid 4 (1,2-di-O-hexadecyl-*sn*-glycero-3-phosphocholine), lipid 5 (1,2-diphytanoyl-sn-glycero-3-phosphocholine), lipid 6 (1,2-di-O-phytanyl-sn-glycero-3-phosphoethanolamine), lipid 7 (1,2-di-O-dodecyl-sn-glycero-3-phosphocholine), lipid 8 (1,2-di-O-octadecyl-sn-glycero-3-phosphocholine), lipid 9 (1,2-di-O-hexyl-sn-glycero-3-phosphocholine), lipid 10 (1,2-di-O-tetradecyl-sn-glycero-3-phosphocholine), and lipid 11 (1,2-di-O- (9Z-octadecenyl)-sn-glycero-3-phosphocholine); see S1 Table] were purchased from Avanti Polar Lipids within Merck. Indium tin oxide (ITO)-coated glass slides were purchased from VisionTek Systems. The fluorescent probe 5(6)-CF, (MW = 376 g/mol), was dissolved in absolute ethanol at a stock concentration of 10 mg/mL. In order to perform all permeability experiments at physiological pH (7.4), the washing buffer was prepared by dissolving sucrose (MW = 342 g/mol) in 5 mM 4-(2-hydroxyethyl)-1-piperazineethanesulfonic acid (HEPES) at pH = 7.4 at a final sucrose concentration of 195 mM. This washing buffer was used for three purposes: (i) to dissolve all tested metabolites; (ii) to electroform vesicles of all lipid chemistries; and (iii) to wash away from the chip the extra-vesicle fluorescent probe CF. All tested metabolites [i.e., glycine (product number G7126 in Merck), alanine (A7627), leucine (L8000), aspartic acid (A9256), glutamine (G8540), tryptophan (T0254), glyceraldehyde (G5001), dihydroxyacetone (PHR1430), deoxyribose (31170), ribose (R7500), arabinose (A3256), glycerol (G9012), cytosine (C3506), uracil (U0750), adenine (A8626), guanine (G11950), urea (U5378), (2-aminoethyl)phosphonic acid (268674), and adenosine monophosphate (A1752); see S2 Table] were dissolved in the washing buffer at a final concentration of 1 mM.

### De novo synthesis of the archaeal 4ME diether G1PC lipid

1,2-di-O-phytanyl-sn-glycero-1-phosphocholine (i.e., 4ME diether G1PC, lipid 1 in S1 Table) was not commercially available during this project and therefore was synthesized de novo by Avanti Polar Lipids based on a previously reported procedure [76] with slight modifications starting with commercially available Solketal. Enquiries about the synthesis and availability of this lipid should be directed to Avanti Polar Lipids.

### Preparation of synthetic lipid vesicles

Giant unilamellar vesicles (simply referred as vesicles in the manuscript) were electroformed by using Vesicle Prep Pro (Nanion) [64]. The phospholipid of interest was weighed and dissolved in chloroform at a stock concentration of 10 mM. Next, 10 μL of each phospholipid

solution was spread evenly using a pipette (Starlab) on the conductive side of an ITO-glass slide. The glass slide was then placed in a vacuum desiccator for 15 min to dry out the lipid solution. Meanwhile, the washing buffer was prepared as described above, and 495 μL of this buffer was degassed and mixed with 5 μL of CF stock solution (10 mg/mL) at a final CF concentration of 0.266 mM. The ITO-slide coated by a dry lipid layer was then placed inside the Vesicle Prep Pro chamber and a greased rubber O-ring was placed around the dry lipid layer. The area within the O-ring was then filled with 300 μL of 0.266 mM CF solution. A second ITO-slide was placed above the O-ring with its conductive side facing downwards and the Vesicle Prep Pro chamber was then closed. The electroformation process was carried out using a different electroformation protocol for each lipid or lipid mixture employed (see S3 Table). Briefly, the electroformation process was performed in three steps. The rise step: The AC voltage was linearly increased peak to peak (p–p) from 0 V till the maximum chosen value (see S3 Table). The main step: The voltage was kept constant for the chosen duration (see S3 Table). The fall step: The voltage was decreased linearly to 0 V. For lipids 9 and 10, none of the protocols employed yielded mechanically stable vesicles. For lipid 6, none of the employed protocols allowed vesicle electroformation possibly because this lipid forms non-lamellar structures of either a cubic or hexagonal fashion [55]. The protocols for the electroformation of vesicles made of lipids 7, 8, and 11 were adapted from [77]. The final fluorescent vesicle suspension consisted of fluorescent vesicles (because of the embedded CF molecules) and free CF molecules in the washing buffer. This suspension was collected from the ITO-slide surface using a pipette, placed in an Eppendorf microcentrifuge tube for storage at 4°C and used within 5 days.

## Extraction of natural lipids

Lipids were extracted from the halophilic *H. volcanii* (100% diether lipids; [48]) and from the thermoacidophilic *S. acidocaldarius* (ca. 10% and 90% diether and tetraether lipids, respectively; [78]). *H. volcanii* DS2 was grown at 45°C (pH 7.2) in Hv-CAB medium containing per liter: 145.0 g NaCl, 17.5 g $MgCl_2 \cdot 6H_2O$, 20.5 g $MgSO_4 \cdot 7H_2O$, 4.1 g KCl, 300.0 mg $CaCl_2 \cdot 2H_2O$, 23.0 mg $FeSO_4 \cdot 7H_2O$, 4.4 mg $ZnSO_4\ 7H_2O$, 3.6 mg $MnCl_2 \cdot 4H_2O$, 0.5 mg $CuSO_4 \cdot 7H_2O$, 1.4 g Tris–HCl, 5.0 g casamino acids, 50.0 mg uracil, 1.0 mg thiamine, and 1.0 mg biotin. *S. acidocaldarius* DSM639 was grown at 75°C (pH 3.0) in Brock medium containing per liter: 1.3 g $(NH_4)_2SO_4$, 280.0 mg $KH_2PO_4$, 250.0 mg $MgSO_4 \cdot 7H_2O$, 70.0 mg $CaCl_2 \cdot 2H_2O$, 20.0 mg $FeCl_3 \cdot 6H_2O$, 4.5 mg $Na_2B_4O_7 \cdot 10H_2O$, 1.8 mg $MnCl_2 \cdot 4H_2O$, 220.0 μg $ZnSO_4 \cdot 7H_2O$, 50 μg $CuCl_2 \cdot 2H_2O$, 30 μg $Na_2MoO_4 \cdot 2H_2O$, 20 μg $VOSO_4 \cdot 2H_2O$, 5 μg $CoSO_4 \cdot 7H_2O$, 1.0 g NZ-amine, and 2.0 g dextrin. Both species were grown with stirring (150 rpm), air bubbling (5 l $min^{-1}$) and 10.0 ml of antifoam 204 (Sigma-Aldrich; Sain-Louis, MO, USA). Cells in late exponential phase were centrifuged (4,000 × *g*, 4°C, 20 min), flash-frozen in $N_{2, liq}$, and freeze-dried overnight. Approximately 2.0 g of dried biomass were extracted for 8 h in uclidehlet extractor containing 400 ml of a 1:9 (v/v) mixture of methanol (MeOH) and dichloromethane (DCM). The resulting total lipid extract was dried under reduced pressure, resuspended in 250 ml of $H_2O$/MeOH (1:1, v/v; F1), and applied onto a Sep-Pak Vac 20 cc C18 cartridge (Waters; Milford, MA, USA). The column was then eluted with 250 ml of $H_2O$/MeOH/DCM (1:2.5:1, v/v/v; F2), 100 ml of $H_2O$/MeOH/DCM (4:25:65, v/v/v; F3), and 100 ml of MeOH/DCM (1:9, v/v; F4). F3 was again dried under reduced pressure, weighted, and stored at −20°C until further use. We made several different attempts to electroform vesicles using these extracted lipids and the following electroformation steps. We employed five different rise steps with a peak-to-peak amplitude of 1, 1.6, 3, 5.3, or 10 V; four different durations for the main step 90, 120, 160, or 360 min. We also explored three different AC frequencies 5, 10, or 500 Hz and four different temperature values 37, 40, 45, or 65°C.

## Design and fabrication of the microfluidic chip

The microfluidic chip was designed in AutoCAD and consisted of two inlets connected to a main chamber that splits into four parallel channels, containing 60 hydrodynamic traps each (henceforth coves) and further connected to a single outlet with an overall footprint of 0.8 mm × 1.3 mm (Fig 1). Each cove was designed by drawing two concentric circles of radii 15 and 35 μm, respectively. A straight vertical line was drawn on the diameter of the outer circle from top to bottom and the left sides of the two semicircles were deleted. A second 10 μm long vertical line was drawn within the inner semicircle, followed by two horizontal semi-infinite lines starting at the vertices of this line. The whole area included in these three lines was then deleted, yielding the final design of each cove. In first approximation, this shape resembles a cove with an opening in the middle. This design was printed on a film mask (Microlithography Services) and used for fabricating a mold of the fluidic chip via photolithography [79,80]. Briefly, a 20-μm thick layer of SU-8 3025 (Microchem) was deposited via spin coating (6,000 r. p.m. for 30 s) on a silicon wafer (Si-Mat Silicon Materials), baked at 95°C for 10 min, exposed to ultraviolet light (1.3 s, 400 nm, 50 mWcm$^{-2}$) through the film mask above, and developed after baking at 65°C for 2 min and 95°C for 3 min. Polydimethylsiloxane (PDMS) replica of this device were produced via soft lithography as previously described [81,82]. Briefly, a 10:1 (base:curing agent) PDMS mixture (SYLGARD 184 Silicone Elastomer Kit, Dow) was cast on the mold, degassed for 30 min, and cured at 70°C for 2 h in an oven. The cured PDMS was peeled from the mold and fluidic accesses were created by using a 1.5-mm biopsy punch (Integra Miltex). Oxygen plasma treatment was used to irreversibly seal the PDMS chip on a glass coverslip (10 s exposure to 30 W plasma power, Plasma etcher, Diener Electronic GmbH). We note that, compared to microfluidic devices previously employed for single-cell confinement [83,84], our approach relies on a single height device that can be fabricated in a single step without the use of dedicated photolithography equipment and can be easily carried out by users who might not be familiar with the technical aspects of microfabrication.

## Microfluidic permeability assay

In the microfluidic chip from the buffer inlet (Fig 1), 15 μL of washing buffer was injected using a pipette until the buffer spilled out from both the metabolite inlet and the outlet. Next, the metabolite inlet was temporarily sealed with a 1 cm × 1 cm piece of tape and 15 μL of the fluorescence vesicle suspension was pipetted into the chip via the washing buffer inlet. Once a cove had been occupied by a vesicle, the amount of fluid entering the cove was reduced [83] but still allowed molecular diffusion in the cove, which was crucial for the permeability assay. Using the tape prevented the fluorescent vesicle suspension from entering into the metabolite inlet. The chip was then transferred under the microscope. A 1-mL plastic syringe (Becton Dickinson) was filled with the washing buffer, and a second 1-mL syringe was filled with a 1-mM solution of the metabolite under investigation. Both syringes were connected to 23-gauge needles (Becton Dickinson) and Masterflex Transfer Tygon tubing with 0.5 mm inner and 1.5 mm outer diameter (Cole-Parmer Instrument). Next, the syringes were connected to a Nemesys pump controlled via the QmixElements software (Centoni). The tape was then removed from the metabolite inlet, and the tubing containing the metabolite solution under investigation was inserted into the metabolite inlet while the metabolite solution was flowing at a rate of 0.5 μL/h. Next, the tubing containing the washing buffer was inserted into the washing buffer inlet while the buffer solution was flowing at a rate of 5 μL/h. These flow rates were then simultaneously increased at steps of 0.5 μL/h every 10 s up to 2 μL/h for the metabolite solution and at steps of 5 μL/h every 10 s up to 25 μL/h for the buffer solution (Fig 1A). These maximal flow rates were kept constant for 20 min to remove any free CF molecules

from the microfluidic environment, while the fluorescent vesicles remained confined in the coves (Fig 1B). Flowing the metabolite solution at a low rate (i.e., 2 μL/h) prevented the accumulation of free CF molecules in the metabolite inlet but did not affect the permeability assay since we obtained similar permeability traits in the absence or presence of this low flow rate. Next, the metabolite solution under investigation was delivered to the fluorescent vesicles in the coves by reducing the flow rate of the washing buffer from 25 μL/h to 1 μL/h and by increasing the flow rate of the metabolite solution from 2 μL/h to 25 μL/h (Fig 1C). Next, the chip was visually inspected by using a 20× 0.85 N.A. oil-immersion objective mounted on an inverted epifluorescence microscope (Olympus IX73) equipped with a sCMOS camera (Zyla 4.2, Andor, used at an exposure time of 0.1s), a blue LED (CoolLED pE300white, used at 30% of its intensity) and a FITC filter. This setup allowed us to simultaneously image 12 coves; therefore, the area of the chip that contained the higher number of vesicles trapped in the coves was chosen. An image of such an area was acquired immediately after increasing the metabolite solution flow rate and then at intervals of 30 s for 3 min. All experiments were performed at an intra- and extra-vesicle pH of 7.4, since pH changes are known to affect CF fluorescence intensity [85]. For each membrane mimic in S1 Table, in order to account for the impact of both the delivery of the washing buffer solution and photobleaching on the intra-vesicle CF fluorescence signal, we performed separate control assays by connecting the metabolite inlet to a syringe containing the washing buffer solution instead of the metabolite solution. Apart from this modification, these control assays were carried out following the protocol described above for the microfluidic permeability assays. The intra-vesicle CF fluorescence consistently linearly decreased during the delivery of the washing buffer for all membrane mimics investigated, and this information was used to provide a background signal for the corresponding microfluidic permeability assays (see Image and data analysis section below).

## Image and data analysis

Images of the fluorescent vesicles for each membrane mimic trapped in the microfluidic coves during the 3-min delivery of each metabolite were imported in ImageJ to produce a temporal image stack [86] and analysed as follows to extract quantitative comparative information about the permeability traits of each membrane mimic to each metabolite. Only images of unilamellar vesicles were retained for analysis, whereas images of multilamellar vesicles were not taken forward. To obtain the single-vesicle temporal dependence of intra-vesicle fluorescence, for each image and each time point, a circle was drawn around each vesicle. A second circle of identical dimensions was drawn in an area 10 μm to the left of the vesicle. This allowed us to extract the mean CF fluorescence of each vesicle and the background around it at each time point, respectively. The background fluorescence was subtracted from the corresponding intra-vesicle fluorescence for each vesicle and each time point. Next, for each vesicle from all the background subtracted values, the initial intra-vesicle fluorescence value (at t = 0) was subtracted. Next, these values were corrected to account for the impact of both the delivery of the washing buffer solution and photobleaching on the intra-vesicle CF fluorescence signal. To do so, a correction factor was calculated from the microfluidic control assay data sets (see below), multiplied by each time value and added to the corresponding intra-vesicle fluorescence value (after the background and initial fluorescence value subtractions above). To obtain a correction factor for each membrane mimic, first, we applied the image analysis protocol above to obtain the single-vesicle temporal dependence of intra-vesicle fluorescence values during the delivery of the washing buffer and subtract from these values the corresponding background and initial intra-vesicle fluorescence value. Second, we averaged these temporal dependences

of corrected single-vesicle fluorescence values to obtain a mean temporal dependence for each membrane mimic during the delivery of the washing buffer. Finally, we fitted this mean temporal dependence to a linear regression with the intercept forced to zero and obtained the slope of the linear fluorescence decrease for each membrane mimic. These slope values were used as correction factors to calculate the permeability of each membrane mimic to each metabolite as described above. For some of the membrane mimic and metabolite pairs, we noticed a minority of outliers (i.e., vesicles that became either much brighter or dimmer with respect to the majority of vesicles investigated during metabolite delivery; see, for example, Fig 2B). This heterogeneity is common when investigating membrane permeability both in vitro [39] and in vivo [38,42]. The permeability of each membrane mimic to each metabolite was measured in three independent experiments from three independent vesicle electroformations. Throughout the manuscript, N indicates the number of single vesicles investigated for each membrane mimic and metabolite pair. N varies across different metabolite experiments investigated since the number of vesicles trapped within the microfluidic coves of the chosen chip area varied. However, care was taken to obtain the same N for each metabolite experiment across the different membrane mimics comparisons to ensure reliable statistical comparisons. In order to do so, when a statistical comparison was to be made between data sets of different N, corrected intra-vesicle fluorescence values were randomly selected from the data set that contained the higher N. Such statistical comparisons were carried out via two-tailed Mann–Whitney tests when comparing distributions of corrected intra-vesicle fluorescence values at t = 3 min for two different lipid types (reported in S2 File) or Kruskal–Wallis one-way analysis of variance when comparing distributions of corrected intra-vesicle fluorescence values at t = 3 min for more than two different lipid types (reported in S2 File). **** indicates a $p$-value $< 0.0001$, *** indicates a $p$-value $< 0.001$, ** indicates a $p$-value $< 0.01$, * indicates a $p$-value $< 0.05$, n.s. indicates a $p$-value $> 0.05$. All data analysis and statistical comparisons were carried out and plotted using GraphPad Prism 9.

## In silico identification of prokaryotic transporter proteins

To characterize the transporter repertoires of bacterial and archaeal species in silico, we first assembled a collection of prokaryotic reference proteomes from UniProt (release 2021_03; [69]). To minimize taxonomic redundancy, a single proteome was selected per genus based on BUSCO completeness scores [87], resulting in 3,044 bacterial and 244 archaeal proteomes. Metagenomes from the bacterial candidate phyla radiation (CPR) were excluded due to their high number and lack of morphological information. To comprehensively identify transporter homologs, profile HMMs derived from TCDB protein families (termed tcDoms, downloaded 2 June 2021) were used to search each proteome using HMMER v3.1b2 ($E < 10^{-5}$, incE $< 10^{-5}$, domE $< 10^{-5}$) [69,88]. If multiple HMMs identified the same predicted protein, the protein was assigned to the family with the lower E-value. To improve the sensitivity of the HMMs, the hits from the initial HMM search were aligned using MAFFT v7.471 (−auto), trimmed with a gap-threshold of 10% using trimAl v1.4, and the resulting alignments were used to generate new HMMs [69,89]. Using the second iteration HMMs, another search was conducted as above, producing the final set of identified proteins. To reduce the potential effects of taxon sampling and dataset bias, prokaryotic taxa were grouped into their respective orders based on NCBI Taxonomy classifications [90]. Transporter abundance was then interpreted as the median number of transporters assigned to a given TCDB family, normalized by the total number of proteins encoded by each taxa, across each order. The resulting distribution was visualized in R v4.0.2, and hierarchical clustering was done using Euclidean distances and the Ward.D2 clustering method implemented by pheatmap (https://mran.microsoft.com/

snapshot/2018-08-31/web/packages/pheatmap/pheatmap.pdf). The differential abundance of individual transporter families was assessed by comparing archaeal and monoderm transporter abundances (given their morphological similarities) using Wilcoxon tests after Bonferroni correction.

To identify putative archaeal transporter families undetected by the TCDB HMMs, we clustered archaeal proteomes into protein families using the Markov clustering algorithm (I = 1.4) based on pairwise BLASTp searches conducted using Diamond v2.0.9.147 (E $< 10^{-5}$, query coverage $> 50\%$, -max-target-seqs $= 10^5$,—more-sensitive search option) [91,92]. Transmembrane (TM) domains were then predicted for each archaeal protein using Phobius, and the median number of TM domains was determined for each protein family [93]. Those families with representation in at least five archaeal species and with a median of at least four TM domains were identified and annotated using eggNOG mapper v2.1.2 [94]. TM domain-containing protein families were classified as putative transporters if they were annotated with PFAM domains associated with transporter function (e.g., all transporter and channel-associated domains) [95]. Bacterial homologs of archaeal protein families were detected using Diamond BLASTp by searching the bacterial reference proteomes with each protein family (E $< 10^{-20}$, query coverage $> 50\%$). To compare TM domain lengths and amino acid composition in Archaea and Bacteria, TM domains were predicted from archaeal and bacterial proteins using Phobius and their length and composition was assessed using BioPython (https://academic.oup.com/bioinformatics/article/25/11/1422/330687?login=true).

## Supporting information

**S1 Fig. Individual archaeal 4ME diether G1PC vesicles are consistently more permeable to a range of metabolites than bacterial diester G3PE-PG-CA vesicles.** Temporal dependence of CF fluorescence in individual archaeal 4ME diether G1PC (magenta dashed lines) and bacterial diester G3PE-PG-CA vesicles (blue dashed-dotted lines) during the exposure to 1 mM of variant metabolites delivered to the microfluidic coves. N is the number of single vesicles investigated for each metabolite exposure and each type of vesicles (magenta and blue for archaeal 4ME diether G1PC and bacterial diester G3PE-PG-CA vesicles, respectively). N varies across different metabolite experiments investigated due to technical constraints (see Methods). However, care has been taken to obtain the same N for each metabolite experiment across the two different types of vesicles to ensure reliable statistical comparisons. The lipids used for creating archaeal 4ME diether G1PC and bacterial diester G3PE-PG-CA vesicles are lipids 1 and 2, respectively, in S1 Table. Numerical values of CF fluorescence in individual vesicles for each lipid type during the delivery of each metabolite are provided in Data A in S1 File. (TIFF)

**S2 Fig. Individual archaeal 4ME diether G1PC vesicles remain more permeable than bacterial diester G3PE-PG-CA vesicles over longer timescales.** Temporal dependence of CF fluorescence in individual archaeal 4ME diether G1PC (magenta dashed lines) and bacterial diester G3PE-PG-CA vesicles (blue dashed-dotted lines) during the exposure to 1 mM of variant glycine, deoxyribose or uracil delivered to the microfluidic coves. Individual archaeal 4ME diether G1PC vesicles remain more permeable than bacterial diester G3PE-PG-CA vesicles over a 6 minute exposure to metabolites. Numerical values of CF fluorescence in individual vesicles for each lipid type during the delivery of each metabolite are provided in Data A in S1 File. (TIFF)

**S3 Fig. Metabolite permeation does not significantly affect vesicle size.** Temporal dependence of the average vesicle size during the delivery of 1 mM deoxyribose (open triangles) or tryptophan (filled triangles) to (**A**) archaeal 4ME diether G1PC vesicles and (**B**) leucine (filled squares) or aspartic acid (open squares) to bacterial diester G3PE-PG-CA vesicles. Mean (symbols) and standard deviation (error bars) were calculated by firstly normalising all values of each vesicle size to the size at t = 0 and then averaging over the values measured on $N$ = 15 vesicles for each metabolite and membrane mimic. Corresponding permeability data for each metabolite and membrane mimic are presented in Fig 2. No significant change in vesicle size was measured during the delivery of deoxyribose or tryptophan to archaeal 4ME diether G1PC vesicles (*p*-value = 0.27 and 0.36, respectively) or during the delivery of aspartic acid or leucine to bacterial diester G3PE-PG-CA vesicles (*p*-value = 0.47 and 0.61, respectively). Moreover, no significant change in vesicle size or shape was measured during the delivery of any of the metabolites in Fig 2. The lipids used for creating archaeal 4ME diether G1PC vesicles and bacterial diester G3PE-PG-CA vesicles are lipids 1 and 2, respectively, in S1 Table. Numerical values of normalized vesicle size for each lipid type during the delivery of each metabolite are provided in Data A in S3 File.
(TIFF)

**S4 Fig. Mono- and ternary-lipid mixtures display similar permeability traits.** Temporal dependence of average CF fluorescence in bacterial vesicles made of one lipid (purple circles) or a ternary lipid mixture (blue squares) during the exposure to 1 mM (**A**) urea, (**B**) glycine, (**C**) glycerol, (**D**) 2-aminoethyl phosphonic acid, (**E**) deoxyribose, or (**F**) ribose delivered to the vesicles at t = 0. Mean (symbols) and standard deviation (error bars) were calculated from at least 10 single-vesicle measurements across 3 independent experiments. N is the number of single vesicles investigated for each metabolite and each type of bacterial vesicle (blue and purple for ternary and single lipid vesicle, respectively). N varies across different metabolite experiments investigated due to technical constraints (see Methods). However, care has been taken to obtain the same N for each metabolite experiment across the two different type of lipid vesicles to ensure reliable statistical comparisons. Such comparisons have been carried out via two-tailed Mann–Whitney tests between the distributions of CF fluorescence values at t = 3 min for the mono- and ternary-lipid mixture. The lipids used for creating the bacterial membrane mimics with ternary and single-lipid mixtures are lipids 2 and 6, respectively, in S1 Table. Numerical values of CF fluorescence in individual vesicles for each lipid type during the delivery of each metabolite are provided in Data B in S1 File.
(TIFF)

**S5 Fig. Lipid chain length does not have a significant impact on the permeability to core metabolites.** Temporal dependence of average CF fluorescence in lipids with tail length of 12 (red circles), 16 (black diamonds) or 18 carbons (blue hexagons) during the exposure to 1 mM (**A**) urea, (**B**) glycine, (**C**) glycerol, (**D**) 2-aminoethyl phosphonic acid, (**E**) deoxyribose, or (**F**) ribose delivered to the vesicles at t = 0. Mean (symbols) and standard deviation (error bars) were calculated from at least 10 single-vesicle measurements (solid lines) across 3 independent experiments. N is the number of single vesicles investigated for each metabolite and each lipid length. N varies across different metabolite experiments investigated due to technical constraints (see Methods). However, care has been taken to obtain the same N for each metabolite experiment across the three different lipid lengths to ensure reliable statistical comparisons. Such comparisons were carried out by evaluating the Pearson correlation coefficient between the average fluorescence value at t = 3 min and the tail length. The measured coefficients were −0.58, 0.95, 0.52, −0.10, −0.69, and 0.30 for urea, glycine, glycerol, 2-aminoethyl phosphonic acid, deoxyribose, and ribose and not statistically significant for any of these metabolites. We

could not investigate the permeability of vesicles with chain length of 14 carbons (lipid 11 in S1 Table) because the transition temperature of these lipids (i.e., 24°C) is very close to room temperature and vesicles easily burst during our permeability assays. Finally, we could not form vesicles using lipids with a chain length of 6 carbons (lipid 10 in S1 Table) despite attempting different electroformation protocols (S3 Table). The lipids used for creating vesicles with tail length of 12, 16, and 18 carbons are lipids 8, 4, and 9, respectively, in S1 Table. Numerical values of CF fluorescence in individual vesicles for each lipid type during the delivery of each metabolite are provided in Data D in S1 File.
(TIFF)

**S6 Fig. Unsaturated lipids favour permeability to urea and glycine.** Temporal dependence of CF fluorescence in the presence (orange squares) or absence of double bonds along the lipid chain (blue circles) during the exposure to 1 mM (**A**) urea, (**B**) glycine, (**C**) glycerol, (**D**) 2-aminoethyl phosphonic acid, (**E**) deoxyribose, or (**F**) ribose delivered to the vesicles at t = 0. Mean (symbols) and standard deviation (error bars) were calculated from at least 10 single-vesicle measurements (solid lines) across 3 independent experiments. N is the number of single vesicles investigated for each metabolite and lipid type. N varies across different metabolite experiments investigated due to technical constraints (see Methods). However, care has been taken to obtain the same N for each metabolite experiment across the two different lipid types to ensure reliable statistical comparisons. Such comparisons have been carried out via two-tailed Mann–Whitney tests between the distributions of CF fluorescence values at t = 3 min for the single and double bonding lipid. The lipids used for creating the archaeal membrane mimics with and without saturation are lipids 9 and 12, respectively, in S1 Table. Numerical values of CF fluorescence in individual vesicles for each lipid type during the delivery of each metabolite are provided in Data E in S1 File.
(TIFF)

**S7 Fig. Archaeal and bacterial transmembrane domains are biochemically consistent. (A)** Heatmap showing transporter repertoires in Archaea and Bacteria, where each row and column represent a prokaryotic order and TCDB transporter family, respectively. Heat map values represent the median number of transporters across each prokaryotic order. The domain (D), group (G), and membrane morphology (M: where blue, purple, and grey represent diderms, monoderms, and unknown membrane morphology, respectively) of each order is noted. TCDB families were grouped by hierarchical clustering based on transporter abundance using Euclidean distances and the Ward.D2 clustering method. Taxonomy was based on NCBI Taxonomy classifications. (**B**) Comparisons between the length of transmembrane domains in Archaea and bacterial monoderms and diderms. Transmembrane domains were identified using Phobius. (**C**) Principal component analyses based on the amino acid compositions of archaeal and bacterial transmembrane domains. In both analyses, archaeal and bacterial transmembrane domains were randomly subsampled to equivalent sample sizes (*n* = 10,000). Numerical values of the data presented are provided at https://doi.org/10.6084/m9.figshare.22086647.
(TIF)

**S1 Table. Synthetic lipids employed to mimic archaeal and bacterial lipid membranes.** List of lipids employed in this work, their chemical structure, their names according to the supplier (Avanti Polar Lipids within Merck), and their molecular weight. We could not produce vesicles using the lipids reported in grey due to technical limitations (see Methods).
(DOCX)

**S2 Table. Physicochemical and permeability properties of all metabolites investigated.** List of metabolites investigated in this work, the class they belong to, their molecular weight (MW), hydrophobicity (decreasing with XLogP3), their charge, the number of their rotatable bonds, the measured average fluorescence of the archaeal or bacterial lipid membrane mimic after 3-min exposure to each metabolite.
(DOCX)

**S3 Table. Protocols to obtain vesicles using synthetic lipids.** Parameters for the protocols employed in this work. The protocols reported in grey yielded a negative outcome (see Methods).
(DOCX)

**S1 File. Numerical values of CF fluorescence during metabolite delivery to vesicles of various lipid types. Data A**. Numerical values of the temporal dependence of CF fluorescence in individual vesicles made of archaeal 4ME diether G1PC lipids or bacterial diester G3PE-PG-CA lipids exposed to glycine, alanine, leucine, aspartic acid, glutamine, tryptophan, glyceraldehyde, dihydroxyacetone, glycerol, deoxyribose, ribose, arabinose, urea, cytosine, uracil, phosphonate, adenine, or guanine. These data are used in Figs 2, S1 and S2. **Data B**. Numerical values of the mean temporal dependence of CF fluorescence in vesicles made of bacterial diester G3PE-PG-CA lipids or bacterial diester G3PE lipids exposed to urea, glycine, glycerol, phosphonate, deoxyribose, and ribose. These data are used in S4 Fig. **Data C**. Numerical values of the mean temporal dependence of CF fluorescence in vesicles made of archaeal 4ME diether G1PC lipids, 4ME diether G3PC lipids, diether G3PC lipids, 4ME Diester G3PC, or bacterial diester G3PE-PG-CA lipids exposed to urea, glycine, glycerol, phosphonate, deoxyribose, and ribose. These data are used in Fig 3. **Data D**. Numerical values of the mean temporal dependence of CF fluorescence in vesicles made of 12:0 Diether G3PC lipids, 16:0 Diether G3PC lipids, or 18:0 Diether G3PC lipids exposed to urea, glycine, glycerol, phosphonate, deoxyribose, and ribose. These data are used in S5 Fig. **Data E**. Numerical values of the mean temporal dependence of CF fluorescence in vesicles made of 18:0 Diether G3PC lipids or 18:1 Diether G3PC lipids exposed to urea, glycine, glycerol, phosphonate, deoxyribose, and ribose. These data are used in S6 Fig.
(XLSX)

**S2 File. Statistical comparisons between permeabilities of two different lipid vesicle types.** Summary of significance and $p$-values estimated using two-tailed Mann–Whitney tests between distributions of CF fluorescence after 3 min of delivery of glycine, alanine, leucine, aspartic acid, glutamine, tryptophan, glyceraldehyde, dihydroxyacetone, glycerol, deoxyribose, ribose, arabinose, urea, cytosine, uracil, phosphonate, adenine, or guanine to individual vesicles made of archaeal 4ME diether G1PC or bacterial diester G3PE-PG-CA lipids.
(XLSX)

**S3 File. Relative change in the size of vesicles of various lipid types during metabolite delivery.** Numerical values of the mean change in the size of vesicles made of archaeal 4ME diether G1PC lipids or bacterial diester G3PE-PG-CA lipids exposed to leucine, aspartic acid, tryptophan, and deoxyribose. These data are used in S3 Fig.
(XLSX)

**S4 File. Statistical comparisons between permeabilities of more than two different lipid vesicle types.** Mean rank difference, summary of significance, and $p$-value estimated using the Kruskal–Wallis one-way analysis of variance test between distributions of CF fluorescence after 3 min of delivery of urea, glycine, ribose, deoxyribose, glycerol, and phosphonate to

individual vesicles made of archaeal 4ME diether G1PC, bacterial diester G3PE-PG-CA, 4ME diether G3PC, 16:0 Diether G3PC, or 4ME 16:0 Diester G3PC lipids.
(XLSX)

**S5 File. Identification of putative archaeal transporter families.** Family: protein family identifier. Sequences: the number of archaeal proteins within a given protein family. Crossover: the percentage of proteins within a given family that were identified by TCDB HMM searches. TM domains: the median number of transmembrane domains per protein across sequences within a protein family. Archaeal_example_seq: a single UniProt accession number of an archaeal protein assigned to a given protein family. Archaeal_taxa: the number of archaeal species encoding proteins assigned to a given family. Bacterial_taxa: the number of bacterial species that contain homologs of a given archaeal protein family based on BLAST comparisons. Putative_transporter: a classification of whether or not a given protein family likely constitutes a transporter based on TCDB overlap and the presence of transporter-associated PFAM domains. Transporter_PFAMs: all PFAM domains assigned to a protein family that were assigned a transporter-associated domain. TopAnnotation: the majority-rule annotation for a given protein family assigned using eggNOG mapper. %withAnnot: the percentage of sequences that were assigned the majority-rule annotation. TotalAnnotSeqs: the number of sequences within the protein family that were successfully annotated. TotalSeqs: the total number of sequences analysed.
(XLSX)

## Author Contributions

**Conceptualization:** Alyson E. Santoro, Thomas A. Richards, Stefano Pagliara.

**Data curation:** Urszula Łapińska, Nicholas A. T. Irwin, Stefano Pagliara.

**Formal analysis:** Urszula Łapińska, Nicholas A. T. Irwin, Thomas A. Richards, Stefano Pagliara.

**Funding acquisition:** Alyson E. Santoro, Thomas A. Richards, Stefano Pagliara.

**Investigation:** Urszula Łapińska, Georgina Glover, Zehra Kahveci, Nicholas A. T. Irwin, David S. Milner, Maxime Tourte, Sonja-Verena Albers, Thomas A. Richards, Stefano Pagliara.

**Methodology:** Urszula Łapińska, Georgina Glover, Zehra Kahveci, Nicholas A. T. Irwin, David S. Milner, Stefano Pagliara.

**Project administration:** Alyson E. Santoro, Thomas A. Richards, Stefano Pagliara.

**Resources:** Maxime Tourte, Sonja-Verena Albers, Thomas A. Richards, Stefano Pagliara.

**Supervision:** Urszula Łapińska, Thomas A. Richards, Stefano Pagliara.

**Validation:** Urszula Łapińska, Thomas A. Richards, Stefano Pagliara.

**Visualization:** Nicholas A. T. Irwin, Thomas A. Richards, Stefano Pagliara.

**Writing – original draft:** Urszula Łapińska, Thomas A. Richards, Stefano Pagliara.

**Writing – review & editing:** Urszula Łapińska, Nicholas A. T. Irwin, Alyson E. Santoro, Thomas A. Richards, Stefano Pagliara.

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
