## [Editor Report · Decision Letter 0]

9 Jan 2023

Dear Tom, 

Happy new year! Thank you for submitting your manuscript entitled "Membrane permeability differentiation at the lipid divide" for consideration as a Research Article by PLOS Biology. Many thanks for your patience while I sought advice over the busy holiday period.

Your manuscript has now been evaluated by the PLOS Biology editorial staff, and an academic editor with relevant expertise has agreed to assess your revisions and responses to the authors (in an ideal world, they'll be able to make a decision themselves. There's a chance that they might ask me to obtain further advice from a specialist, but we'll cross that bridge when we come to it...)

However, whichever route we end up taking, we need you to complete your submission by providing the metadata that is required for full assessment. To this end, please login to Editorial Manager where you will find the paper in the 'Submissions Needing Revisions' folder on your homepage. Please click 'Revise Submission' from the Action Links and complete all additional questions in the submission questionnaire.

Once your full submission is complete, your paper will undergo a series of checks in preparation for the next step. To provide the metadata for your submission, please Login to Editorial Manager (https://www.editorialmanager.com/pbiology) within two working days, i.e. by Jan 11 2023 11:59PM.

Best wishes,

Roli

Roland Roberts, PhD

Senior Editor

PLOS Biology

rroberts@plos.org

---

## [Editor Report · Decision Letter 1]

8 Feb 2023

Dear Tom,

Thank you for your patience while your manuscript "Membrane permeability differentiation at the lipid divide" was assessed at PLOS Biology. It has now been evaluated by the PLOS Biology editors, and an Academic Editor with relevant expertise. 

Based on our Academic Editor's assessment of your revisions and your responses to the prior PNAS reviews, we are likely to accept this manuscript for publication, provided you satisfactorily address the remaining points raised by the Academic Editor (see foot of this email), and the following data and other policy-related requests.

IMPORTANT. Please attend to the following:

a) Please attend to the comments from the Academic Editor that can be found at the foot of this email. The Academic Editor (whose identity will be fairly self-evident) made it clear that while these are not mandatory, they might be advisable.

b) The current Title will be very opaque to anyone outside the immediate field, and we ask that you change it to something much more explicit and informative. We suggest "Systematic comparison of unilamellar vesicles reveals that archaeal lipid membranes are more permeable than bacterial ones."

c) Please address my Data Policy requests below; specifically, we need you to supply the numerical values underlying Figs 2ABCDEFGHIJKLMNOPQR, 3ABCDEFG, 4AB, S1ABCDEFGHIJKLMNOPQR, S2ABC, S3AB, S4ABCDEF, S5ABCDEF, S6ABCDEF, S7ABC, either as a supplementary data file or as a permanent DOI’d deposition.

d) Please cite the location of the data clearly in all relevant main and supplementary Figure legends, e.g. “The data underlying this Figure can be found in S1 Data” or “The data underlying this Figure can be found in https://doi.org/XXXX”

We expect to receive your revised manuscript within two weeks. 

*Published Peer Review History*

*Press*

Sincerely,

Roli

Roland Roberts, PhD

Senior Editor,

rroberts@plos.org,

PLOS Biology

DATA POLICY:

Regardless of the method selected, please ensure that you provide the individual numerical values that underlie the summary data displayed in the following figure panels as they are essential for readers to assess your analysis and to reproduce it: Figs 2ABCDEFGHIJKLMNOPQR, 3ABCDEFG, 4AB, S1ABCDEFGHIJKLMNOPQR, S2ABC, S3AB, S4ABCDEF, S5ABCDEF, S6ABCDEF, S7ABC. NOTE: the numerical data provided should include all replicates AND the way in which the plotted mean and errors were derived (it should not present only the mean/average values).

DATA NOT SHOWN?

ACADEMIC EDITOR'S COMMENTS:

1. On page 2 line 70 Introduction) the authors write:

“An additional study has shown that, in the absence of counter-ions, liposomes made of diether lipids extracted from archaea (e.g. from Halobacterium salinarum) display lower permeability to protons compared to liposomes made of lipids extracted from bacteria”.

I do not think it is possible to conduct a study without counter-ions: H+ has a charge and that charge must be balanced by something. That might be a Cl- ion. If an H+ crosses a membrane a positive charge moves into the cell, and that positive charge will oppose the influx of other H+ ions into the cell. If a Cl- also crosses into the cell, that positive charge will be partially dispersed by the negative charge. The rate at which H+ moves in therefore ultimately depends on the permeability of the membrane to Cl-, not H+. A counter-ion does not mean a positive charge moving the other way, but an ion that counters the build-up of charge, and that would include Cl-. It is therefore a counter-ion. If you used SO42- instead of Cl- and it crossed the membrane more slowly (which it probably would) then you would measure a lower permeability to protons. I have not read the papers cited but either they did not say what is imputed to them, or if they did then the authors did not know what they were talking about. Either way, this should be clarified for the authors not to look foolish to any bioenergeticist.

2. At the bottom of page 4 the authors write:

“These data suggest that the metabolic selectivity of these membrane mimics is complex and does not rely solely on basic molecular properties, such as molecular weight or hydrophobicity.”

As an aside, we have also found some strange patterns between hydrophobicity and binding interactions in Harrison et al (A biophysical basis for the emergence of the genetic code in protocells. BBA Bioenerg. 2022 1863: 148597)… It seems that a more subtle measure such as partition energy, discussed by in Caldararo & di Giulio (The genetic code is very close to a global optimum in a model of its origin taking into account both the partition energy of amino acids and their biosynthetic relationships, Biosystems 214 (2022), 104613). Some brief discussion of partition energy in terms of properties of metabolites crossing the membrane might be helpful.

3. Page 9 bottom of page:

The authors write:

“Regardless, transporter reduction may have been facilitated, in part, by the increased metabolite permeability of archaeal lipid membranes which, in turn, reduces the requirement for, or utility of, transporter proteins.”

This argument is interesting and may be true. But another possible explanation is simply differential loss of membrane proteins since then, as they are especially labile. See for example Sojo et al (Membrane Proteins Are Dramatically Less Conserved than Water-Soluble Proteins across the Tree of Life. MBE 33 (2016) 2874–2884). If, for example, archaea were more likely to be autotrophic than bacteria, then there may have been a greater tendency to lose membrane transporters as there was less need for them. Such a possibility might at least be mentioned.

4. Conclusions (page 10):

These are now admirably concise and give very little perspective. That might be a virtue, though a few extra words of context might not go amiss. I had previously specifically raised questions about the likelihood of the common ancestor of archaea being a heterotroph, and there are good recent papers from Williams et al and others on LACA and LBCA (which also imply autotrophic ancestry. Does this paper imply that archaea were heterotrophs, lived by recycling in biofilms, were autotrophs, or what? A few words that tentatively point to the significance of the paper in relation to what we think we might know from phylogenetics might not go amiss. Some of this was raised in response to the referees and could possibly be incorporated into the paper itself. I don’t need to agree; I would like them to offer something, even if couched tentatively.

---

## [Editor Report · Decision Letter 2]

22 Feb 2023

Dear Tom,

Thank you for the submission of your revised Research Article "Systematic comparison of unilamellar vesicles reveals that archaeal core lipid membranes are more permeable than bacterial ones" for publication in PLOS Biology. On behalf of my colleagues and the Academic Editor, Nick Lane, I'm pleased to say that we can in principle accept your manuscript for publication, provided you address any remaining formatting and reporting issues. These will be detailed in an email you should receive within 2-3 business days from our colleagues in the journal operations team; no action is required from you until then. Please note that we will not be able to formally accept your manuscript and schedule it for publication until you have completed any requested changes.

IMPORTANT: Many thanks for making the requested changes to the manuscript and supplying data underlying the main Figs. However, I couldn't find the data underlying the Supplementary Figs (specifically, S1ABCDEFGHIJKLMNOPQR, S2ABC, S3AB, S4ABCDEF, S5ABCDEF, S6ABCDEF, S7ABC), so I'll be asking my colleagues to pass on a request to you (alongside their additional requests) to supply these and to cite the location of the data in the respective legends.

Sincerely, 

Roli

Senior Editor

PLOS Biology

rroberts@plos.org